# Transfer Learning for Automated Anterograde Tracer Signal Segmentation in Marmoset Brain Microscopy Images

**Muhammad Febrian Rachmadi**[1]          FEBRIAN.RACHMADI@RIKEN.JP
**Akiya Watakabe**[2]          AKIYA.WATAKABE@RIKEN.JP
**Tetsuo Yamamori**[2]          TETSUO.YAMAMORI@RIKEN.JP
**Henrik Skibbe**[1]          HENRIK.SKIBBE@RIKEN.JP

[1] *Brain Image Analysis Unit, RIKEN Center for Brain Science, Wako-shi, Japan*

[2] *Molecular Analysis of Higher Brain Function, RIKEN Center for Brain Science, Wako-shi, Japan*

**Editors:** Under Review for MIDL 2021

## Abstract

A goal of contemporary neuroscience research is to map the structural connectivity of primate brains using microscopy imaging data. In this context, the Japan Brain/MINDS project aims at using neural tracers to map neural connections in the marmoset brain. As part of the analysis, automated segmentation of tracer signals in microscopy images is demanded. We process two kinds of tracer images based on retrograde neural tracers and anterograde tracer images. While retrogradely connected cell bodies can be manually annotated quite easily in a reasonable time, the annotation of entire anterograde tracer images is impractical. We found that with transfer learning, a small amount of training data is sufficient to adapt networks that have been trained for the detection of retrogradely cell bodies to successfully segment cells and axonal projections in anterograde tracer images. In this paper, we explain the methodology and promising preliminary results.

**Keywords:** Brain microscopy images, neuron tracer segmentation, transfer learning.

## 1. Background and Motivation

One of neuroscience research's goals is to map the structural connectivity of the brain using tracer images. Tracer image is obtained by injecting fluorescent agents to an individual and staining sections of the brain using antibody of the fluorescent agents. The infected neurons will show (i.e., glow) in different colours based on specific fluorescent agents used. There are generally two types of tracer images based on how the neurons are infected, which are anterograde and retrograde tracer images (Lanciego and Wouterlood, 2020). Anterograde tracer images show the cell bodies of neurons in the injection site along with their axonal connections projecting to the other parts of the brain. Anterograde tracer images are usually used to trace neural connection from their source to their point of termination. Retrograde tracer images can be used to observe the opposite kind of connections. They show the cell bodies of neurons in parts of the brain that are connecting to cells in the injection site. A major difference in the appearance is that we only see the cell bodies as circular blobs in retrograde tracer images - axons are not visible. That makes manual annotation easy.

The motivation of this study is to develop an automated system that can perform automatic segmentation of anterograde tracer signals given manual labels of retrogradely connected cell bodies in marmoset brain microscopy images. While retrogradely connected cell bodies can be manually annotated relatively easy in a reasonable time, the annotation

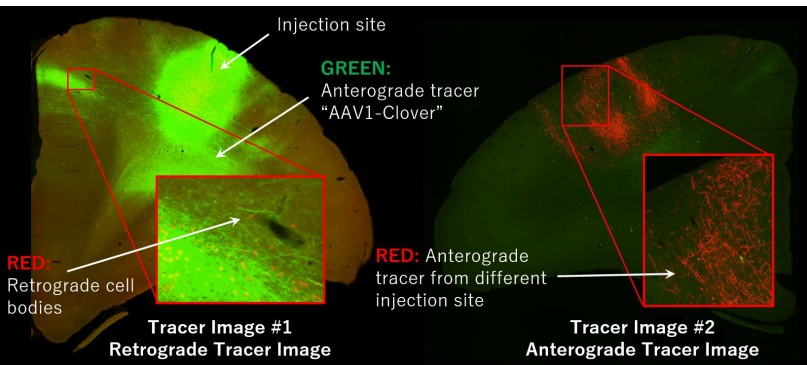

Figure 1: Retrograde and Anterograde Tracer Images

of entire anterograde tracer images is impractical. In this study, the shape of retrogradely connected cell bodies is similar for most of them whereas anterograde tracer signals have different sizes and shapes following the structure of neuron's axons. Figure 1 shows the visual differences between the retrogradely connected cell bodies (i.e., shown in red colour in retrograde tracer image (left)) and anterograde axonal connections (i.e., shown in red in anterograde tracer image (right)). It is noteworthy that there is second anterograde tracer signal in green in both retrograde and anterograde tracer images; but this green signal is not the focus of this study.

## 2. Proposed Approach and Preliminary Result

In this study, we investigated an approach in which a trained network (i.e., U-Net (Ronneberger et al., 2015)) for automatic detection of cell bodies in retrograde images is repurposed for the segmentation of anterograde tracer signals via transfer learning (i.e., task adaptation). This approach is similar to our previous study where we repurposed a U-Net for calculating irregularity map for white matter hyperintensities segmentation (Rachmadi et al., 2018). As previously mentioned, it is relatively easier to manually label retrogradely connected cell bodies in retrograde images. We successfully managed to manually label 445 retrograde images of brain section from 11 subjects (of marmoset). Each retrogradely connected cell body is manually labeled by an expert as a dot sized $2 \times 2$ regardless of its size or shape for simplicity.

The U-Net was firstly trained in a supervised manner for detection of retrogradely connected cell bodies (by segmenting the dots of cell bodies) in a set of training patches (sized $512 \times 512$) randomly sampled from 380 retrograde images (roughly about 20,000 training patches) while the rest of the retrograde images were used for testing. The U-Net was trained for 32 epochs with Adam optimization, 0.001 learning rate, and early stop.

After that, only three examples of semi-automatically annotated anterograde tracer images of marmoset brain sections where sufficient to exploit the trained U-Net for anterograde tracer signal segmentation. The anterograde labels were obtained by cleaning the raw anterograde tracer signal that was associated with the red color channel using morphology operations (e.g., sharpening, contrast enhancement, dilation, erosion). In the retraining, we used 16 epochs, with Adam optimization, 0.00001 learning rate, and early stop.

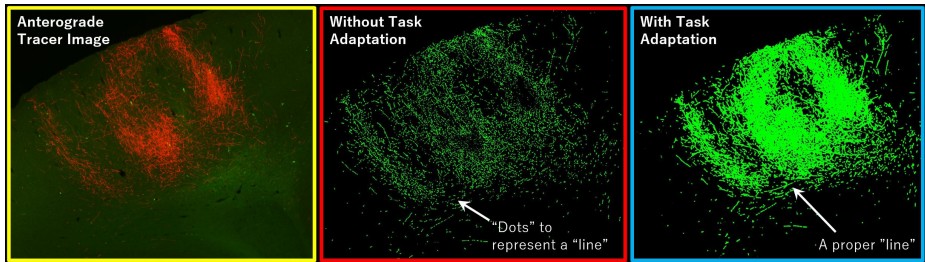

Figure 2: Anterograde Tracer Segmentation with and without Transfer Learning

Figure 2 shows the segmentation results before (red box) and after the (blue box) transfer learning process (i.e., retraining or task adaptation). We can see that before transfer learning, the U-Net was not able to correctly segment the elongated structure of the axons in the anterograde signal. This is not surprising, because it has been trained to detect small circular cells rather than axons. After the task adaptation, however, the U-Net showed a very promising performance on anterograde tracer signals.

## 3. Challenge and Future Work

One challenge of the current approach is that anterograde tracer images can vary greatly due to different green and red intensities. The greenish anterograde images shown in Figures 1 and 2 can be regarded as "champion" data (i.e., anterograde tracer signals are shown in a clear red colour with green tissues in the background). Because of that, task adaptation itself might not be sufficient. We are no exploring domain adaptation and image enhancing techniques to further improve the results.

## Acknowledgments

This research was supported by the program for Brain Mapping by Integrated Neurotechnologies for Disease Studies (Brain/MINDS) from the Japan Agency for Medical Research and Development AMED (JP21dm0207001).

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
