# OpenReview forum: "Transfer Learning for Automated Anterograde Tracer Signal Segmentation in Marmoset Brain Microscopy Images"
_MIDL.io/2021/Conference/Short — Submitted to MIDL 2021_

### Official Review · Reviewer_GMFp · 2021-04-30

**Confidence:** 4
**Final Rating:** 3

**Summary:**

The authors process two kinds of tracer images based on retrograde neural tracers and anterograde tracer images. The authors use a small amount of training data is sufficient to adapt networks that have been trained for the detection of retrogradely cell bodies to successfully segment cells and axonal projections in anterograde tracer images.

**Strengths:**

The paper is well-written.  The strengths are:
1. Transfer learning is used to reduce training data for a new domain/task.
2. Nice application in neuroscience.
3. Experimental results are well explained.


**Weaknesses:**

There are a few weaknesses from a technical view, but maybe not totally weaknesses as it is a short paper.
1. Did the author use any data augmentation during the training/fine-tuning stage? Data augmentation could be helpful.
2. Loss function for the training is not mentioned. The structure looks like a kind of tabular. Maybe a new loss function [1] would be helpful to improve the segmentation results where the Dice loss or cross-entropy loss may have some drawbacks. Please see [1] for details.
3. Quantitative results are missing. Some evaluation metrics, such as Hausdorff distance, Dice score could be used to quantify and monitor the accuracy.



[1] clDice - a Novel Topology-Preserving Loss Function for Tubular Structure Segmentation, CVPR 2021

**Deanonymize Review:**

no

**Detailed Comments:**

Mainly from the weakness part:

1. Did the author use any data augmentation during the training/fine-tuning stage? Data augmentation could be helpful and further reduce the training data.
2. Loss function for the training is not mentioned. The structure looks like a kind of tabular. Maybe a new loss function [1] would be helpful to improve the segmentation results where the Dice loss or cross-entropy loss may have some drawbacks. Please see [1] for details.
3. Quantitative results are missing. Some evaluation metrics, such as Hausdorff distance, Dice score could used to quantify and monitor the accuracy.
4. Image translation to perform domain adaptation could be helpful too.


**Justification Of The Rating:**

1. Transfer learning is used to reduce training data for a new domain/task.
2. Nice application in neuroscience and the experimental results are well explained.
3. Results of quantitative metrics are missing.

**Paper Type:**

validation/application paper

**Special Issue:**

no

---

### Official Review · Reviewer_MFwD · 2021-05-07

**Confidence:** 4
**Final Rating:** 2

**Summary:**

Authors propose a transfer learning approach for automatic anterograde tracer signal segmentation. Their method leverages a trained U-Net for retrograde segmentation, which is argued to be an easier problem. This network is repurposed for anterograde segmentation. Author present qualitative results for visual evaluation of the performance of their approach.

**Strengths:**

- Article is well-written and it is easy to understand.
- Authors demonstrate good knowledge on domain specific information related to how anterograde and retrograde tracers produce contrast in the image.

**Weaknesses:**

- The significance of the work is not clear.
- Experimental setup is highly limited.
- Provided results are not convincing. There is only a single figure for qualitative inspection, and no quantitative evaluations are provided.
- There is no comparison made against a baseline approach such as simple thresholding of the red channel.

**Deanonymize Review:**

no

**Detailed Comments:**

The manuscript is well-written. The organization of the text is logical and the use of language is adequate. I recommend the authors to expand figure captions and provide more information. Figure captions are typically expected to provide enough information to understand the displayed content in the presented image. I believe readers will appreciate the information provided about the different working mechanisms and contrasts obtained in the two tracer injection methods. However, for this work, it would be expected to provide information about state-of-the-art approaches that solve this problem - and what lead the authors to design their own method instead of using an existing technique.

**Justification Of The Rating:**

The motivation behind why to use a sophisticated deep learning approach is not justified. Authors have not mentioned about any competing approaches. For example, it is not clear why simple thresholding would not work to solve the mentioned problem. Methodologically, the proposed approach is not very interesting. The provided experiments and results are not convincing. Authors only show a single image for qualitative inspection.

**Paper Type:**

validation/application paper

**Special Issue:**

no

---

### Meta-Review · Program_Chairs · 2021-05-09

**Recommendation:** Reject
**Confidence:** 4

**Metareview:**

The contribution of the paper is to apply a well-known deep learning approach to the problem of anterograde tracer images. For such paper, some more quantitative evaluation would be needed.

---

### Decision · Program_Chairs · 2021-05-11

Reject